# Elo Method and Race Traits: A New Integrated System for Sport Horse Genetic Evaluation

**DOI:** 10.3390/ani10071145

**Published:** 2020-07-06

**Authors:** Andrea Giontella, Francesca Maria Sarti, Giovanni Paolo Biggio, Samira Giovannini, Raffaele Cherchi, Maurizio Silvestrelli, Camillo Pieramati

**Affiliations:** 1Department of Veterinary Medicine—Sportive Horse Research Center, University of Perugia, via S.Costanzo 4, 06123 Perugia, Italy; maurizio.silvestrelli@unipg.it (M.S.); camillo.pieramati@unipg.it (C.P.); 2Department of Agricultural, Food and Environmental Sciences, University of Perugia, Borgo XX Giugno, 74, 06121 Perugia, Italy; francesca.sarti@unipg.it (F.M.S.); samira.giovannini@gmail.com (S.G.); 3AGRIS, Servizio Ricerca Qualità e Valorizzazione delle Produzioni Equine, piazza D. Borgia, 4, 07014 Ozieri, Sassari, Italy; giampaolo.biggio@gmail.com (G.P.B.); ilex283@gmail.com (R.C.)

**Keywords:** Sardinian Anglo-Arab horse, race traits, Elo method, genetic evaluation

## Abstract

**Simple Summary:**

The current selection of the Sardinian Anglo-Arab horse (SAA) for racing is not based on any scientific method because the breeders decide only on their knowledge of pedigree and on the comparison of racing results. The comparison between competitors is a common situation in sports and games, and it might cause problems in horse performance evaluation. To overcome this problem in the game of chess, Elo suggested using a method based on winning chances. Elo’s system assumed that ranking probability for two competitors could be estimated from their rating difference. Even in horse races, the ranking of each horse, and consequently its earnings, depends on the level of its competitors. A genetic index based on Elo’s rating can clearly show the value of each animal. The strength of this method is that it allows evaluation of a horse by considering various different traits such as wins, placings, earnings over the entire career altogether.

**Abstract:**

This first survey on Sardinian Anglo-Arab horse (SAA) race traits highlights important aspects for the breeding purpose of this population. The heritability of the race traits were estimated through a trivariate model; the estimates were 0.39, 0.33, and 0.30 for the number of placings, total earnings and Elo rating, respectively. The genetic progress could be improved by using an MT genetic evaluation of stallions and mares, combining information from competition traits.

## 1. Introduction

The Sardinian Anglo-Arab (SAA) is a very popular breed for equestrian sports, originating from crosses between old Arabian and part-Arab lines and Thoroughbreds [1]. In Italy, most of the foals come from Sardinia, which is world-famous for horse breeding. Although the Italian Studbook of this breed was established in 1967, the official information on pedigree in Sardinia traces back to 1874, when an army remount station was established in Ozieri (50 Km SE from Sassari) [2]. The SAA horse is widely used in flat races in France as well as Italy, and the running distances range from 1000 m to 2400 m. Nowadays in Italy, approximately 160 SAA horses participate in about 150 races (around 10 horses per race) each year, and the first four places are awarded prizes.

In the racing competitions, the SAA horses compete only against individuals of the same breed. In the equestrian competitions, the SAA horses also compete against horses of other breeds; there are no specific events only for SAA in jumping or endurance. SAA mares are frequently crossbred with European warmbloods to obtain more competitive horses in jumping, or with Arabian stallions, to obtain more competitive horses in endurance [3].

The current selection of SAA for racing is not based on any scientific method: the breeders only rely on the survey of the pedigree and on the comparison of racing results. The trade in live stallions and frozen semen across countries allows greater access to superior breeding stock, and this activity is an important part of the equine industry economy in many countries [4]. Therefore, SAA breeders need specific tools to help them select stallions correctly because they spend large sums of money buying stallions and semen. An effective breeding program needs to be developed through close cooperation between scientists and breeders to obtain an adequate tool to operate a suitable selection of stallions and mares [5].

In other racing breeds, such as Thoroughbred and trotter horse, several studies have aimed to estimate the heritability of the “performance” measured by different traits [6,7,8,9].

Speed seems to be a very important trait to evaluate horses competing in a race, and it has been widely used in trotters, although transformed into the classical “time per km (or per mile)” [10]. Other common traits are the placings and the annual or career earnings. Usually, these traits are not normally distributed, therefore, some kind of transformation, like logarithm or square root, is applied [6,7,8,11,12,13].

A further problem with both placings and earnings is that the ranking of each horse, and consequently, its earnings, depends on the level of its competitors [14]. The comparison between competitors is a common situation in sports and games, and this also occurs with jumping horses [15]. To overcome these problems in the game of chess, a method based on winning chances was suggested [16]. Elo’s system assumed every performance to be a normally distributed random variable, so that ranking probability for two competitors can be estimated from their rating difference. In the academic world, Thurstone had already studied the estimation of hidden variables determining observed rankings in psychometrics [17]. In recent years, both Elo-like methods and the Thurstonian model have been used to estimate breeding values of horses from competition data [12,18,19,20].

This study aimed to evaluate SAA estimated breeding values (EBVs), taking into consideration both career, traits such as wins, placings as well as earnings and Elo estimated racing performance.

## 2. Material and Methods

### 2.1. Pedigree File

All the ancestors of the recorded horses were added to the pedigree file for the genetic evaluation, including at least four generations for each participating horse, giving a total of 43,624 horses (27,052 females and 16,572 males) born from 1853 to 2013. The genealogical information was provided by the Agricultural Research Agency of Sardinia - Istituto Incremento Ippico della Sardegna (AGRIS). The pedigree file, from the first to the fifth generation, showed a completeness of 86.44%, 79.87%, 75.82%, 72.88% and 68.53%. The mean of the maximum number of traced generations reported in the Studbook was equal to 11.15. The average inbreeding coefficient (F) was 0.012 [2].

### 2.2. Stat Dataset

Ten different traits for 1756 SAA born between 1995 and 2006 (ended career) were calculated using a database of career-aggregated statistics provided by H.I.D. (Hippo Information Technologies and Data S.p.A.). These traits, referred to the total career (more than 1 individual record/year), were divided in two groups: “result traits” (number of wins, number of placings, wins per start and placings per start), and “economic traits” (total earnings, logarithm of total earnings, square root of total earnings in career, earnings per start, logarithm of earnings per start and, square root of earning per start).

The transformation of both total earnings and total earnings per start into their logarithm and square root was used to better fit a normal distribution. Since earnings data were collected in a short period (ten years), they were not further modified to account for prize development over time. Descriptive statistics were computed on the ten traits and on the effects with R software [21].

The estimation of genetic parameters was performed on the “results traits” (4 × 4) and the “economic traits” (6 × 6) using MTDFREML software [22] following a multivariate BLUP Animal Model using the univariate estimates of single-trait models for each trait as priors.

The basic model was:Y = Xb + Zu + e(1)
where:

y = vector of observations of the corresponding trait;

b = vector of fixed effects: sex organized in 3 levels (males, females, geldings) and year of birth divided into 12 levels (from 1995 to 2006);

u = vector of individual additive genetic values;

e = vector of residual error;

X and Z = incidence matrices, fixed and random effects, respectively.

The random-effects have the following distribution:(2)$$var\left(\begin{array}{c} a\\ e \end{array}\right)=\left[\begin{array}{cc} A\;\sigma_{a}^{2} & 0\\ 0 & I\sigma_{e}^{2} \end{array}\right]$$
where A is the additive relationship matrix, I an identity matrix; *σ_a_²* the additive genetic variance and *σ_e_²* the error variance.

The estimates of heritabilities (h²) and genetic correlation (r_g_) from the two MT models were used to choose a “result trait” out of four and an “economic trait” out of six, so that overall response, that is the result of the direct response (i h^2^_i_ σ_i_) and all the indirect responses (i r_gij_ h_i_ h_j_ σ_j_), was maximized within each group of traits. For any mass selection of intensity *i*. the overall standardized selection response of each trait*_i_* was calculated as:(3)$$h_{i}^{2}+\sum r_{g_{ij}}h_{i}h_{j}$$
where *j* is any other trait in the “stat” dataset [23].

### 2.3. “Elo” Dataset

“Elo” dataset consists of 15,898 result records referred to 1974 horses in 2139 races between 1992 and 2013. These data, provided by H.I.D, were used to reconstruct the ranking of each race. An Elo method was used to estimate the rating of each horse. In its first race, each horse was assumed to have a rating of 2000 points and to perform randomly with a standard deviation of 200 points. The ranking of each horse at the end of a race matched the ranking of its performance in comparison to the performance of the other participants within the same race. For each participant in a race, the difference between its expected ranking from the ratings of the others and the actual observed ranking at the end of the race, was used to adjust upward or downward the rating using the formula:new rating = old rating + *k* × (observed ranking—expected ranking)(4)

Many sports and games use integer values in the 10 to 40 range for “*k*”, the update coefficient in the previous formula. In order to choose an appropriate value for the update coefficient, an iterative procedure was set up, using all “*k*” values from 0.5 to 200 by a 0.5 step; for each value of “*k*” in this range, the ratings of all horses in any race were calculated by their previous results, and the best-rated horse was predicted as the winner of the next race. The “*k*” value was set to 15 in further analyses because it showed the best ability of predicting the winner (34.1% correct predictions).

The estimates of Elo ratings were performed by in-house Fortran 95 software.

The final rating of each horse (whole career) was analyzed in a univariate model with no fixed effect because no statistical significance were obtained for sex and year of birth. The descriptive statistic (R project) [21] on Elo final rating effects are reported in Appendix A.

### 2.4. Final Dataset

In order to reduce the number of sets of indexes, a final dataset was prepared by merging the “Elo” dataset with the number of placings and the square root of total earnings in a career that were found to be the traits maximizing the overall selection response in the “stat”.

The final dataset included 2360 horse records (1370 horses with 3 traits are from both dataset “Elo” and “stat”, 604 only from the “Elo” final rating, 386 only from “stat” dataset). This dataset was analyzed in an MT animal model, including the fixed effects of sex and year of birth only for the two “stat” traits. The accuracy values were examined to show the improvement of using the overall genetic index evaluation in comparison to mass selection. In more detail, the number of animals without records that showed an accuracy greater than the minimum observed value in the animals with records was used for the comparison.

Variance components estimation and their standard errors were performed by VCE package [24]. Estimates of the EBVs and their accuracies were performed by MTDFREML [22]. A Spearman rank correlation was performed with R software [21] between the EBVs estimated by simple trait and EBVs estimated by MT.

## 3. Results

### Stat, Elo and Final Dataset

Descriptive statistic results (computed with R project [21]) for “stat” traits and Elo rating effects are reported respectively in Table 1 and Appendix A.

The h^2^ estimates by univariate models on the “stat” dataset and Elo rating (Table 2) ranged from a minimum of 0.19 (wins per start) to a maximum of 0.48 (number of placings) for the “result” traits and from a minimum of 0.31 (earnings per start) to a maximum of 0.42 (square root of total earnings) for the “economic” traits.

Although the logarithmic transformation is probably the most used method to make the earnings traits quasi-normally distributed, the square root transformation showed a higher heritability in this study. The Elo rating h^2^ was equal to 0.29.

The MT model estimations of heritability and correlations for the 4 “result traits” are shown in Table 3.

The h^2^ were higher than those calculated in the univariate model, with the exceptions of placings per start which decreased from 0.37 to 0.34, and of the number of placings which stayed on 0.48; all the genetic correlations were equal to or higher than 0.7. Correlations close to 1 were found for the number of wins and the number of placings and for wins per starts and placings per starts; these higher correlations could have an effect on the genetic parameter estimates (collinearity), and this could help to explain the differences among single trait and multi-trait analyses.

The environmental correlations were much lower than the genetic correlations, with the only exception of the correlation between the number of placings and the number of wins (0.81). The estimates of heritability and correlations by MT model for the six “economic” traits are reported in Table 4.

Unlike the “result” traits, a slight reduction in the estimate of h² was observed with respect to the univariate model, except for the total earnings, which slightly increased from 0.35 to 0.36. Genetic correlations of 0.95 or higher occurred among original traits and the respective derived traits (e.g., total earnings and logarithm and the square root of total earnings) and the environmental correlations were higher than those of the “result” traits. A general view of the above-mentioned correlations, in particular the genetic ones, can lead to consider that in practice, some traits are none other than the logarithm or square root of the same trait. High genetic correlations also occur between total earnings and the earning per start (0.81), and this justifies the use of just one of these traits for the following analysis with Elo.

In the final trivariate model (Table 5), the h^2^ values of the number of placings and the square root of total earnings decreased from 0.48 to 0.39 and from 0.42 to 0.33, respectively, whilst the Elo rating increased to 0.30. Both the genetic and environmental correlations between these three traits were high: mainly those between “stat” traits.

In Table 6, the different efficacy of mass selection and selection index is shown.

In the univariate model for the number of placings, 166 animals without records were estimated with more accuracy than the least accurate animal with records; in the univariate model for the square root of total earnings and for Elo rating, these animals were 255 and 2354, respectively. The number of animals without records with an EBV as accurate as that of animals with records was impressively increased in the final trivariate model. These animals were 9156 for the number of placings, 4646 for the square root of total earnings and 3465 for Elo rating; the animals without records that were more accurate for all the three EBVs were 3457. As expected, the accuracy reaches higher values (0.51–0.54) in the trivariate model than in the univariate model (0.31–0.38). The Spearman ranking correlation, between the EBVs estimated by univariate and EBVs estimated by MT, showed coefficients higher than 0.95 (*p* ≥0.05, data not shown).

## 4. Discussion

### Stat, Elo and Final Dataset

It is well known that if the measure of the performance is the racing time or any other result at the individual race level, then many effects must be considered such as racing distance, racing track, meteorological conditions, etc.; therefore, the recommended method is summarizing all these effects into an “individual race” effect [25]. This “individual race” effect takes into consideration the value of the competitors, avoiding the subjective grouping-levelling of races; however, it does not include other important effects such as sex, age, handicap, or jockey. The heritability of racing time in Arabian horses was estimated between 0.17 ± 0.001 [26] and 0.28 ± 0.003 [27].

If a career trait is used as a performance measurement, the number of effects that must be considered is reduced by a considerable amount. In this research, eleven (ten traits form “stat” plus Elo) performance traits were considered, and ten of them were career traits: the effects of sex and year of birth were included in the animal model used to get estimates.

Furthermore, since the traits were also averaged per start, the number of races was considered. In other research, this effect was fitted in the model as a covariate [28]. It must be noted that both methods probably reduce the importance of horse longevity-durability. Two different transformations have been fitted to the total earnings: the logarithmic [7,8] and the square root transformation [6,7,8]. The last considered trait, the “Elo” performance, was a measure at individual race level and it should correct two different problems at the same time: the normality of the performance distribution and the different level of the competitors. Furthermore, most horses finish out of placings and money in every race; therefore, neither the results traits nor the economic traits measure the differences between these horses: the drawback is that the real importance of the measured differences between these horses, for instance, the last and the second-to-last one, is questionable.

No estimation was available for heritability of result traits in the Arabian horse: the most comparable traits seemed to be ranking where estimates were 0.25 [29], 0.18 [30], 0.12 ± 0.01 [31] and 0.07 ± 0.01 [27]. More estimates were available for heritability of earnings traits in the same breed: 0.22 [29], 0.19 [30], 0.09 ± 0.01 (Log-transformed) [31], 0.17 ± 0.017 (earnings per start) [27], 0.17 ± 0.04 (Log earnings per start) [27] and 0.46 ± 0.15 [26]. Only one genetic correlation between “result” and “earning” traits, namely ranking and Log earnings [30] has been previously computed in the Arabian horse: the estimate was very high (0.97 ± 0.01) and similar to the values (0.94–0.97) found for Thoroughbred in steeplechase [32]. Elo rating was estimated only in trotter horses, and the heritability of this trait was 0.27 ± 0.08 [18]. In another trotter population, a Thurstonian model, which is a competitive evaluation similar to Elo method, provided a lower estimate of 0.09 ± 0.02 [12]; likewise in Thoroughbred horses, the estimate was 0.047 ± 0.014 [33] and in other breeds range from 0.07 to 0.17 [34]. Furthermore, since all the correlations were very high, the trait to choose for maximizing the overall response was the most heritable trait, which was a further advantage. In the final MT model, the genetic correlations between the three traits were medium-high. That allows a breeder to plan a breeding program in a way that is less difficult, and the selection is streamlined.

## 5. Conclusions

Although breeders can have different goals in the selection, since high genetic correlations between “stat” traits were obtained, they are sure that, by choosing just one trait in each group, indirect selection will, nevertheless, bring a general improvement for that group of traits.

Nowadays, the breeders decide only on their knowledge of pedigree and racing results; a key step is to convince them that an EBV of a horse without record can be more accurate than that of a racing horse. Trusting in the trivariate model, the number of horses that a breeder can consider for reproduction greatly increases and this is a further benefit for the SAA population. The calculation of an official Elo rating could show the value of each animal to the horse fans in an easy and understandable way: this could help in making comparisons between horses so that new bettors are attracted. It is reasonable to expect that, with the integration of an Elo rating in the genetic evaluation system, the breeders could benefit from the same clarity; therefore, Elo ratings could have a great outlook in sport horse breeding.

## Figures and Tables

**Table 1 animals-10-01145-t001:** Descriptive statistic results for the ten traits referred to the total career (more than 1 individual record/year).

Traits										
Descriptive Statistic	1	2	3	4	5	6	7	8	9	10
mean	1.26	5.04	0.07	0.32	5496.75	2.31	47.47	300.38	9.77	1.24
s.d.	2.54	7.91	0.012	0.28	12,488.4	1.72	56.97	647.39	337.54	338.13
cv	201.88	156.96	161.25	87.61	227.2	74.3	120.01	215.53	3455.15	27,233.6
min.	0	0	0	0	0	0	0	0	0	0
max.	27	76	1	1	157,438	5.2	396.78	7323.6	3.86	85.58

1 = number of win, 2 = number of placings, 3 = wins per start, 4 = placings per start, 5 = total earnings, 6 = logarithm of total earnings, 7 = square root of total earnings, 8 = earnings per start, 9 = logarithm of earnings per start, 10 = square root of earnings per start.

**Table 2 animals-10-01145-t002:** Estimates of animal effect ± s.e., residual effect ± s.e., heritability ± s.e. from univariate models.

Trait	Animal Effect	Residual Effect	$\mathrm{Heritability}\;(\sigma_{a}^{2}/\sigma_{p}^{2})$
*Result*			
number of wins	2.36 ± 0.45	4.25 ± 0.34	0.36 ± 0.06
number of placings	31.89 ± 5.02	34.24 ± 3.68	0.48 ± 0.06
wins per start	0.0027 ± 0.0008	0.0110 ± 0.0007	0.19 ±0.05
placings per start	0.032 ± 0.007	0.054 ± 0.005	0.37 ± 0.07
*Economic*			
total earnings	56 × 10^6^ ± 10 × 10^6^	105 × 10^6^ ± 8 × 10^6^	0.35 ± 0.06
logarithm of total earnings	1.06 ± 0.25	2.06 ± 0.19	0.34 ± 0.07
square root of total earnings	1437 ± 236	1966 ± 172	0.42 ± 0.06
earnings per start	95.152 ± 20.549	214.072 ± 16.083	0.31 ± 0.06
logarithm of earnings per start	0.51 ± 0.13	1.03 ± 0.09	0.33 ± 0.07
square root of earnings per start	62.4 ± 11.5	97.71 ± 8.4	0.39 ± 0.06
Elo rating	2993.5 ± 509.7	7488.8 ± 427.1	0.29 ± 0.04

**Table 3 animals-10-01145-t003:** Estimates of “result” traits heritability ± s.e. (diagonal), genetic (above diagonal) and environmental (below diagonal) correlation ± s.e. from multiple trait model.

Trait	Number of Wins	Number of Placings	Wins per Start	Placings per Start
number of wins	**0.41 ± 0.05**	0.99 ± 0.01	0.82 ± 0.06	0.79 ± 0.07
number of placings	0.81 ± 0.02	**0.48 ± 0.05**	0.73 ± 0.08	0.70 ± 0.08
wins per start	0.58 ± 0.03	0.33 ± 0.05	**0.25 ± 0.05**	1.00 ± 0.00
placings per start	0.36 ± 0.04	0.54 ± 0.04	0.46 ± 0.03	**0.34 ± 0.05**

**Table 4 animals-10-01145-t004:** Estimates of “Economic” traits heritability ± s.e. (diagonal), genetic (above diagonal), and environmental (below diagonal) correlation ± s.e. from multiple trait model.

Traits	1	2	3	4	5	6
**1** total earnings	**0.36 ± 0.02**	0.96 ± 0.04	0.99 ± 0.01	0.81 ± 0.03	0.93 ± 0.05	0.89 ± 0.03
**2** logarithm of total earnings	0.35 ± 0.02	**0.24 ± 0.02**	0.99 ± 0.02	0.89 ± 0.03	0.99 ± 0.00	0.96 ± 0.01
**3** square root of total earnings	0.85 ± 0.01	0.73 ± 0.01	**0.41 ± 0.02**	0.85 ± 0.03	0.96 ± 0.03	0.93 ± 0.01
**4** earnings per start	0.69 ± 0.02	0.46 ± 0.02	0.74 ± 0.02	**0.29 ± 0.02**	0.95 ± 0.02	0.99 ± 0.01
**5** logarithm of earnings per start	0.32 ± 0.02	0.99 ± 0.00	0.69 ± 0.01	0.49 ± 0.02	**0.23 ± 0.02**	0.99 ± 0.00
**6** square root of earnings per start	0.59 ± 0.02	0.84 ± 0.01	0.85 ± 0.01	0.84 ± 0.01	0.86 ± 0.01	**0.34 ± 0.02**

**Table 5 animals-10-01145-t005:** Estimates heritability ± s.e. (diagonal), genetic (above diagonal) and environmental (below diagonal) correlation ± s.e. from the final multiple trait model.

Trait	Number of Placing Square	Root of Total Earning	Elo Rating
number of placing	**0.39 ± 0.04**	0.90 ± 0.03	0.65 ± 0.08
square root of total earning	0.85 ± 0.01	**0.33 ± 0.04**	0.87 ± 0.04
Elo rating	0.58 ± 0.03	0.67 ± 0.02	**0.30 ± 0.04**

**Table 6 animals-10-01145-t006:** Number and accuracy of the animals without records showing accuracy equal or higher than that animals with records.

Trait	Model			
	Univariate		Trivariate	
	n	Accuracy	n	Accuracy
number of placing	166	0.38	9156	0.51
square root of total earning	255	0.31	4646	0.54
Elo rating	2354	0.30	3465	0.50

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
