# Peer review of "Elo Method and Race Traits: A New Integrated System for Sport Horse Genetic Evaluation"

_animals, 2020, doi:10.3390/ani10071145_

Round 1

Reviewer 1 Report

Although the manuscript is a little bit improved but still I am not satisfied with the statistical models they used yet. They should either use the birth year (not divided by 12 or age) as fixed effect. Age is more accurate to me. Also, since the animals are coming from different private sectors, those should be used in the models too since the management and diet and environment is different. Otherwise, you model is not correct as it is right now.

The presented mode is not correct too as they used the same Z for both random additive genetic effects and permanent environment effects!

Also, the reason for transformation is not clear to me if it didn't change the normality of the data.

Reviewer 2 Report

I accept the changes.

Author Response

We thank the Referee for the attention and collaboration.

Reviewer 3 Report

The study “Elo method and race traits: a new integrated system for sport horse genetic evaluation” suggests the use of an adaptation of the ELO system used to classify the chess players for the genetic evaluation of competition horses. The paper is well written and interesting for publication in ANIMALS. However, I have some comments or suggestions that need to be addressed before acceptance.

Line 65. I think the authors should refer the original paper of the Thurstonian approach (Gianola and Simianer, 2006; Genetics 174: 1613-1624)

Line 93. The model does not include the age of the horse. Authors must justify why they are not included in the model. Also, the rider and horse-rider interaction are generally included as random effects. Their absence may imply an increase of the heritability estimates.

Line 108. I do not understand the meaning of this formula. The overall response in a multiple trait framework can be increased by improving the accuracy of each trait, not adding the direct with the correlated response. It is irrelevant for the paper. Therefore, I suggest removing it.

Line 121. The authors must include information of  the criteria to select the “k” parameter.

Line 125. The ELO rating of a horse varied throughout its life. Therefore, a horse’s final rating does not exactly reflect the horse’s performance as old horses must be surely penalized. I think this can be solved by including an age effect or taking the best ELO rating throughout the life of the horses rather the final one.

Line 152. It will interesting to present descriptive statistics of the ELO ratings.

Line 290. The thurstonian method considers the performance of the horses in each competition and not a single value throughout their lives.

Line 300. Please, delete the “6”.

Line 361. There are missing references.

Round 2

Reviewer 1 Report

Still, I am not convinced by their statistical models. I think their models need to be reassesed based on the information. The information is not very well defined to understand what models are the best fit. 

Author Response

As requested we respond to the Academic Editor

Reviewer 3 Report

The authors responded appropriately to all my comments or suggestions.

Author Response

As requested we respond to the Academic Editor

This manuscript is a resubmission of an earlier submission. The following is a list of the peer review reports and author responses from that submission.

Round 1

Reviewer 1 Report

The manuscript "Elo method and race traits: a new integrated system for sport horse genetic evaluation" written by Giontella et al. investigated the potential of Elo's method for selection of Sardinian Anglo-Arab horses and estimated the heritabilities and genetic correlations among a few traits in this population. The study seems to be interesting for horse breeders and horse racing fans but the methods employed by authors didn't follow the correct path or wasn't explained clearly and I wouldn't recommend this for publication. 

Major Comments:

L25: Why square root of total earnings? I assumed they have done some transformation but they didn't need to mention the transformed here. Just the trait is enough.

L25-17: The authors described the importance of Elo's method in the abstract and this study and they should describe their findings using Elo's method here.

What do you mean by selection results? I suggest estimating the genetic trend and genetic gain (key equation) over the years.

Introduction: 

Most of the references are not very new except for two of them that haven't been published yet (under review). I suggest adding the newer references in the introduction too.

L 69-72: Why did you use the transformation and the original trait as separate trait. You should only present transformed trait. Here the reason of transforming is missing as you reported both of them. I assumed that you used the transformation as your original trait was not normally distributed. Right? If yes, you didn't need to report the original data as they didn't follow the normal distribution.

L73: One transformation is enough. Why the authors used to do two transformation methods? They should clarify it or either present the best one.

L74: Better normality? What do you mean by better? Did original data follow the normal dist. or not? We don't have better or worse?

L78-81: Which model did you use for estimating h2 and rg? It is not clear to me which models they used? Which fixed or random effects they included in their models? They should bring a table showing the significance of each fixed effect and random effect affecting each trait.

It is not clear to me whether they used the same model across all traits?

Did they use year or age as fixed effect?

Also, information about the feed and nutrition of animals, where the animals were grown, which feed, water, they have access to... etc is missing. A lot of information is missing in the data part that must be added to the manuscript.

In the results, they mentioned the models, which they should bring it to the M&M with detailed information.

Pedigree information is needed. How many generations did you use?

L78: They should mention which h2 estimates they reported? Were they the average of multi-variate estimates or univariate results?

L82: Each component of the model must be defined.

L101: Why no fixed effect? Don't you think that age, gender, herd has effect on the rating?

L106: The number of animals are very small to estimate the genetic parameters for them especially 604 and 386 and this will affect the parameters estimation.

L108-111: Not clear. What is rT1?

Results:

One table of all effects including fixed and random effects for each trait must be included.

Also, Descriptive statistics of all traits are needed including mean, CV, range, etc.

Table 1: 

The authors should only look at one-way of transformation that followed the normal distribution and not those that they didn't follow the normal distribution. The authors should also explained the normality of all traits in the study.

Table 1: With the small sample size of 608, this is wired that you had a small SE? How do you explain this? I am doubt that with 608 records you can get this small SE.

L147: I am doubt that we have genetic correlation of 1? How do you justify this? Did all of the analyses were converged? This might be due to not considering the correct model.

L148: Phenotypic correlations are normally higher than genetic correlations but here phenotypic correlations are lower than rg. How do you explain this?

Table 3: This table basically shows just two traits with different transformation and as you can see all the results are pretty much similar. It doesn't make sense to me to present both transformations and one trait. The best transformation should only be presented.

L159-160: This is obvious that they should be similar but the exception is almost the same.

Table 4: Still, I am not clear how did you come up with these three traits. More explanation is needed in M&M to clarify this.

Table 5: rT1 is not a symbol for the accuracy of breeding values. I suggest using the accuracy of EBV estimation instead throughout the manuscript.

Change to tri-variate and Single trait  or univariate

The estimated accuracies are very low. How come that the number of animals are different between tri-variate and univariate. This is not clear. You should only report the number of records of certain trait not the record of all traits and pedigree animals. This is not correct.

Minor Comments:

L14: Delete "on"

L17: differences

L26: What is MT? You should define it first.

L37: 1,000 m to 2,400 m.

L53. I wouldn't say these are the problems.

L60: I wouldn't call these years recent years.

L66: All the numbers have a space instead of comma. Correct them throughout the manuscript.

L77: Do you mean Multi-trait model? You should define it first

L85: delete "result"

L86: Change "these data" to "that"

L113: This should be defined first not here!

L139: Four instead of 4

L140. dot at the end.

L151: which was decreased

L151: higher than 0.69

L154: six instead of six

L191: There is no need for this as you don't have any other sub-seciton

Reviewer 2 Report

The authors should present the pedigree structure of the analyzed horses.

Model of variance component estimation and BLUP prediction should be described in detail in methodology.

VCE package calculates BLUP solutions and their standard errors. There is no need to use package MTDFREML.

Evaluation and interpretation of the Spearman rank correlation would be a valuable element in the study.